



# ESR-thermochronometry of the Hida range of the Japanese Alps: Validation and future potential

Georgina E. King[1], Sumiko Tsukamoto[2], Frédéric Herman[1], Rabiul H. Biswas[1], Shigeru Sueoka[3] and Takahiro Tagami[4]

[1]Institute of Earth Surface Dynamics, University of Lausanne, Lausanne, Switzerland
[2]Leibniz Institute for Applied Geophysics, Hannover, Germany
[3]Tono Geoscience Center, Japan Atomic Energy Agency, Toki, Japan
[4]Division of Earth and Planetary Sciences, Kyoto University, Kyoto, Japan

*Correspondence to*: Georgina E. King (georgina.king@unil.ch)

**Abstract.** The electron spin resonance (ESR) of quartz has previously been shown to have potential for determining rock cooling histories, however this technique remains underdeveloped. In this study, we explore the ESR of a suite of samples from the Hida range of the Japanese Alps. We develop measurement protocols and models to constrain the natural trapped charge concentration as well as the parameters that govern signal growth and signal thermal decay. The thermal stability of the Al and Ti-centres is similar to that of the luminescence of feldspar. Inverting the ESR data for cooling yields similar thermal histories to paired luminescence data from the same samples. However, a series of synthetic inversions show that whereas the luminescence of feldspar can only resolve minimum cooling histories of ~160 °C/Myr over timescales of $10^{3-5}$ a, quartz ESR may resolve cooling histories as low as 25-50 °C/Myr over timescales of $10^{3-7}$ a. This difference arises because quartz ESR has a higher dating limit than the luminescence of feldspar. These results imply that quartz ESR will be widely applicable in the constraint of late-stage rock cooling histories, providing new insights into landscape evolution over late-Quaternary timescales.

## 1. Introduction

Thermochronometry based on trapped-charge dating allows the constraint of late stage exhumation and/or rock thermal histories at the scale of glacial-interglacial cycles (e.g. Biswas et al., 2018). Following the study of Herman et al. (2010) which applied optically stimulated luminescence (OSL) dating to constrain the exhumation histories of the Southern Alps of New Zealand, there have been a number of both methodological and applied studies that have almost exclusively focused on luminescence dating (see King et al., 2016a and Herman and King, 2018 for reviews). In this study we explore the potential of a second trapped-charge dating method, electron spin resonance (ESR) of quartz, for ultra-low temperature (i.e. < 100°C) thermochronometry.

Electron spin resonance can be used to measure the time-dependent accumulation of unpaired electrons (paramagnetic centres) in minerals such as quartz (cf. Grün, 1989; Ikeya, 1993). As for luminescence dating, when a mineral is exposed to ionizing radiation, electrons are excited from their ground state in the valence band, to the conduction band. Almost immediately most electrons fall back to the valence band, recombining with the "holes" of positive charge created by the electron's excitation.





However, some electrons become trapped within defects in the crystal lattice, caused by element vacancies or substitutions. In this study we specifically target the Al (hole trapping) centre and the Ti (electron trapping) centre, although other defects such as the E' (oxygen vacancy) centre could also be investigated (e.g. Grün et al., 1999). The Al-centre comprises a hole located at $AlO^-$ (Nuttall and Weil, 1981) whereas the Ti-centre comprises the substitution of $Si^{4+}$ with $Ti^{3+}$+ $e^-$ stabilized with $H^+$, $Li^+$

or $Na^+$ (Rinneberg and Weil, 1972; Isoya et al., 1983). ESR offers a key advantage over luminescence dating, specifically that ESR signals saturate later (Rink, 1997; Tsukamoto et al., 2018). Within the context of thermochronometry, this means that whilst the application of luminescence thermochronometry remains geographically limited to regions experiencing extremely rapid cooling/exhumation higher than several mm/yr e.g. New Zealand (Herman et al., 2010), eastern Himalayan syntaxis (King et al., 2016b), ESR thermochronometry could be much more widely applied.

The potential of ESR for thermochronometry has been recognized previously. Following from an earlier study (Ikeya, 1983), Toyoda and Ikeya (1991) first suggested that the intensity of quartz ESR centres could be used to determine the low-temperature thermal histories of the host rock. Scherer et al. (1993; 1994) investigated changing ESR centre intensities with depth through the known-thermal history KTB borehole in Germany (Coyle et al., 1997), which has also been used to validate

the luminescence thermochronometry technique (Guralnik et al., 2015, Biswas et al., 2018). Scherer et al. (1993; 1994) recorded a qualitative reduction in signal intensity of the Al-centre with increasing temperature and depth. In contrast, data for the Ti-centre were much more scattered with zero signal intensity recorded for many samples. However, it was Grün et al. (1999) who reported the first quantitative ESR-thermochronometry results from their study of the Eldzhurtinskiy Granite from the Russian Caucasus. Using the Al and Ti-centres of quartz, they obtained cooling rates of between 160 °C/Myr and 600

°C/Myr which correspond to denudation rates of ~2.5 and 5.5 mm/yr.

Despite the potential illustrated by ESR-thermochronometry in these early studies, the technique has not been applied since, in part associated with the difficulties of making ESR measurements (i.e. gamma or X-ray source availability, absence of automated instrumentation). In this study, we investigate the potential of ESR-thermochronometry through applying new

measurement protocols (Tsukamoto et al., 2015), which have been facilitated by developments in instrumentation (Oppermann and Tsukamoto, 2015), and that have recently been validated against samples with independent age control (Richter et al., In Press). We propose a kinetic model inspired by recent progress in luminescence thermochronometry (Lambert et al., In Review) to facilitate the inference of rock thermal histories from ESR laboratory data and perform a series of synthetic inversions to evaluate the range of cooling histories that ESR thermochronometry may be applicable over. We then investigate

six rock samples from the Japanese Alps and contrast their ESR thermal histories with those obtained from paired optically stimulated luminescence thermochronometry of feldspar (e.g. Guralnik et al., 2015; King et al., 2016b,c) of the same samples.





## 2. Theoretical basis

The theoretical basis of ESR-thermochronometry is very similar to that of luminescence thermochronometry (cf. King et al., 2016a; Herman and King, 2018 for reviews), with the advantage that unlike feldspar minerals, quartz minerals are not thought to suffer from athermal signal losses. Here we present the kinetic model for ESR-thermochronometry, before discussing how the parameters that describe signal growth and signal thermal decay can be constrained in the laboratory.

### 2.1 Kinetic model

We propose the following kinetic models to describe the evolution of ESR signals with temperature. A saturating system may be described by:

$$\frac{\partial[\tilde{n}(E_a,t)]}{\partial t} = \widetilde{D}[1 - \tilde{n}(E_a,t)] - s\, e^{-\frac{E_a-\mu(E_t)}{k_B T}}[\tilde{n}(E_a,t)] \tag{1}$$

and a non-saturating system can be described by:

$$\frac{\partial[\tilde{n}(E_a,t)]}{\partial t} = \widetilde{D}[\tilde{n}(E_a,t)] - s\, e^{-\frac{E_a-\mu(E_t)}{k_B T}}[\tilde{n}(E_a,t)] \tag{2}$$

where,

$$\tilde{n}(t) = \int_{E_a=0}^{\infty} P(E_a)\,\tilde{n}(E_a,t)\,dE_a \tag{3}$$

and,

$$P(E_a) = \frac{1}{\sigma(E_t)\sqrt{2\pi}}\exp\left(-\frac{1}{2}\left(\frac{E_a-\mu(E_t)}{\sigma(E_t)}\right)^2\right). \tag{4}$$

where $\tilde{n}$ is the trapped charge population with activation energy, $E_a$ (eV). In the instance of a saturating system $\tilde{n}$ is expressed as a saturation ratio, but for a non-saturating system it is expressed as absorbed radiation dose (Gy). The first term on the right-hand side of Eqs. (1) and (2) describes charge trapping as a first-order process. For a non-saturating system, $\widetilde{D}$ is defined by the environmental dose rate $\dot{D}$ (Gy), whereas for a saturating system, $\widetilde{D}$ is defined as $\dot{D}/D_0$ where $D_0$ is the characteristic dose of saturation (Gy). The second term on the right-hand side of Eqs. (1) and (2) describes thermal charge detrapping, and here we benefit from recent advances made in luminescence thermochronometry, and follow Lambert et al. (In Review) by describing thermal detrapping using a model that assumes a Gaussian distribution of activation energies, $E_a$ around the mean



trap depth, $\mu(E_t)$ (eV). Thermal detrapping is also described by the frequency factor, $s$ (s⁻¹), the Boltzman constant, $k_B$ (eV), temperature, $T$ (K) and $P(E_a)$ the probability of thermally evicting electrons (or holes) from the trap (Eq. (4)).

## 2.2 Constraining charge trapping

The natural trapped charge concentration, which reflects the equilibrium between charge trapping and thermally stimulated charge detrapping, can be measured in the laboratory through the development of a sample specific radiation dose response curve. This comprises measurement of a sample following increasingly large laboratory radiation doses, and interpolation of the natural ESR signal onto the resultant dose response curve. Measurements can either be made on single (e.g. Tsukamoto et al., 2015) or using multiple aliquots of the same sample (e.g. Grün et al., 1999). The former approach has only recently been made practical, following the introduction of X-ray irradiation for regenerative dosing (Oppermann and Tsukamoto, 2015).

## 2.3 Constraining charge detrapping

Thermal detrapping can be measured following laboratory isothermal decay experiments, whereby aliquots of a sample are given a radiation dose before being heated at different temperatures for different durations. The resultant signal loss is measured and fitted with the kinetic model described in Eqs. (1-4). Previous investigations have suggested that the thermal decay of quartz ESR can be described by first order or second order kinetics. Here, instead we use a density of states model, originally developed for the luminescence of feldspar (Li and Li, 2013; Lambert et al., In Review; further details of model selection are given in the Supplementary Material). The selected model is based on a Gaussian distribution of activation energies $\sigma(E_t)$, around the mean trap-depth, $\mu(E_t)$ (Lambert et al., In Review), and may be applicable for quartz ESR data where electrons can be trapped in a variety of different defects, e.g. $Ti^{3+} + e^-$ charge compensated by $H^+$, $Li^+$ or $Na^+$ (Tsukamoto et al., 2018).

## 3. Assessing the potential of ESR-thermochronometry

Electron spin resonance dating analyses are not automated, meaning that the laboratory measurements required for ESR-thermochronometry analyses are considerably more time-consuming than those required for luminescence thermochronometry. It is thus necessary to verify that ESR-thermochronometry offers advantages over luminescence methods. To achieve this, a series of synthetic inversions for known cooling histories were done using the kinetic parameters of sample KRG16-06 (Table 1).

## 3.1 Forward modelling

Five different monotonic cooling scenarios were used to test the potential of ESR-thermochronometry in comparison to OSL-thermochronometry, comprising cooling with rates of 100 °C/Myr, 75 °C/Myr, 50 °C/Myr, 25 °C/Myr and no cooling (i.e. isothermal holding at 0 °C for 2 Myr). All cooling rates were maintained for at least 2 Myr with a starting temperature of 200 °C which is greater than the anticipated closure temperature of the ESR system (cf. Grün et al., 1999; Scherer, 1993; 1994).





Using the kinetic model in Eqs. (1) and (2), a trapped charge population, $\tilde{n}_{fwd}$ was predicted for both the Ti and Al-centres respectively using the kinetic parameters of sample KRG16-06 (Table 1) for the five different scenarios. In addition, the same exercise was carried out for four feldspar multi-OSL-thermochronometry signals of the same sample using the following kinetic model, after King et al., 2016a (see supplementary information for further details on model selection):

$$\frac{d[\tilde{n}(r',E_b,t)]}{dt} = \widetilde{D}[1 - \tilde{n}(r',E_a,t)] - s\,e^{-\frac{E_t-E_b}{k_BT}}[\tilde{n}(r',E_b,t)] - \tilde{s}e^{-\rho'^{-\frac{1}{3}}r'}[\tilde{n}(r',E_b,t)] \qquad (5)$$

where the total accumulation of charge with time, i.e. $\tilde{n}(t)$ is obtained by integrating $\tilde{n}(r',E_b,t)$ over the range of band-tail states, $E_b$, and an infinite range of dimensionless distances, $r'$:

$$\tilde{n}(t) = \int_{r'=0}^{\infty}\int_{E_b=0}^{E_t} p(r')P(E_b)\,\tilde{n}(r',E_b,t)\,dE_b\,dr' \qquad (6)$$

where $P(E_b)$ is the probability of evicting electrons into band-tail states of energy $E_b + dE_b$, defined as:

$$P(E_b) = B\,e\left(-\frac{E_b}{E_u}\right)dE_b \qquad (7)$$

where $B$ is a pre-exponential multiplier, and where $p(r')$ is the probability density distribution of the nearest recombination centre defined by Huntley (2006) as:

$$p(r')dr = 3r'^2 e^{-r'^3}dr' \qquad (8)$$

where dimensionless distance $r' \equiv \left\{\frac{4\pi\rho}{3}\right\}^{\frac{1}{3}} r$, the dimensionless density of recombination centres $\rho' \equiv \frac{4\pi\rho}{3\alpha^3}$ and $\alpha$ is a constant (Huntley, 2006; Kars et al., 2008; Tachiya and Mozumder, 1974).


### 3.2 Inverse modelling

We inverted the five sets of $\tilde{n}_{fwd}$ values for the ESR and OSL data described above using a similar approach to King et al. (2016a), which we briefly outline here. The trapped-charge (or hole) populations were modelled for 10,000 randomly generated time-temperature histories (t-T paths), which were constrained to cool monotonically between 200 °C and 0±5 °C, over 2 Ma.

We computed the dose response curves by solving the differential equations described above using a semi implicit Euler



method (Press, 2007). For each t-T path we calculated a misfit between the inverted trapped-charge population $\tilde{n}_{mod}$ and our forward modelled values $\tilde{n}_{fwd}$ (Wheelock et al., 2015), from which the misfit, $M$ and likelihood, $L$ are calculated:

$$M = \Sigma_1^m \left( 0.5 \frac{\tilde{n}_{fwd}}{\sigma} \left( \log\left(\frac{\tilde{n}_{fwd}}{\tilde{n}_{mod}}\right) \right) \right)^2 \tag{9}$$

$$L = \exp(-M) \tag{10}$$

for $m$ traps, where $\sigma$ is the uncertainty. An arbitrary uncertainty on $\tilde{n}_{fwd}$ of 10% was assumed. Cooling histories are then accepted or rejected by contrasting $L$ with a random number between 0 and 1; if $L$ is greater, the cooling history is retained.
The accepted cooling histories are finally combined to construct a time-temperature history probability density function through dividing the time-temperature axis into 50 intervals and summing the number of paths that cross through each of the different cells. The Al, Ti and OSL data were first inverted separately and then the Al and Ti-centres were inverted together.

The results of the forward modelling and the synthetic inversions for the ESR and OSL data are shown in Fig. 1. The OSL
signals for all cooling histories reach saturation (Fig. 1c), and this is reflected in the failure of the OSL to recover any of the cooling histories when inverted. The minimum cooling rate that can be resolved using OSL for sample KRG16-06 is ~160 °C/Myr, calculated from 86% of the luminescence signal saturation level. Signal saturation is the key limitation that restricts the application of luminescence thermochronometry to regions undergoing rapid exhumation. In contrast, it is clear that the ESR data are able to resolve the 100 °C/Myr, 75 °C/Myr and 50 °C/Myr synthetic cooling histories clearly, and cooling rates
of 25 °C/Myr are distinct from isothermal holding at 0 °C over timescales of ~2 Ma. These results are significant as they show that ESR-thermochronometry is applicable in a range of geological settings beyond the rapidly exhuming locations that luminescence-thermochronometry is currently restricted to.

**4. Proof of concept – Hida range, Japanese Alps**

To further explore the potential of the ESR method we applied it to a suite of samples from the Hida range of the Japanese Alps. The Japanese Alps which reach elevations of up to 3,000 m are thought to have uplifted since the Pliocene or Quaternary (Yonekura et al., 2001; Tokahashi, 2006) in response to E-W compressional tectonic forces (Takahashi, 2006; Townend and Zoback, 2006; Sueoka et al., 2016). Lithology of the Hida range is dominated by granitic intrusions, including the Kurobegawa
granite, which is the youngest known intrusion on Earth and which was emplaced between 10-0.8 Ma ago (Ito et al., 2013; 2017). Previous efforts to apply apatite fission-track dating on the Kurobegawa granite have been unsuccessful because of the



very low fission-track density (Yamada, 1999). Extremely young apatite ($0.50 \pm 0.04$ Ma) and zircon helium ages ($0.37 \pm 0.10$ Ma) have recently been reported (Spencer et al., 2019), indicating that exhumation in this region has remained rapid throughout the Quaternary Period.

5 Six bedrock samples were taken from the Kurobegawa granite, northern Hida range of the Japanese Alps. Four surface samples were taken and form an elevation transect, whilst a further two samples were taken from a high-temperature tunnel, which has a present-day temperature of ~40-50 °C but which had temperatures of up to 165 °C at the time of excavation in the late 1930s (Yuhara and Yamamoto, 1983). Samples had a minimum size of 15 x 15 x 15 cm, to ensure that a light safe portion could be extracted from their interiors. Sample details are given in the Supplementary Material.

## 10 4.1 Sample preparation

Bedrock samples were prepared using standard laboratory methods under subdued red light conditions at the University of Lausanne and University of Bern, Switzerland (cf. King et al., 2016c). At least 10 mm was cut from the exterior of the samples using a water-cooled diamond saw, to extract the light safe interior. A thin section was made using a representative sample of the bedrock exterior and a further representative sample was sent to ActLabs, Canada for ICP-MS analysis. Sample interiors 15 were then hand crushed to extract the 180-212 μm grain size fraction, which was treated with HCl and $H_2O_2$ to remove any carbonates and organic material respectively. The K-feldspar and quartz fractions were separated from heavy minerals using heavy liquids. The K-feldspars were retained for luminescence dating, whilst the quartz extracts ($2.58 > \rho < 2.70$ g cm$^{-3}$) were etched for 40 minutes using 40% HF, before being treated with HCl to remove fluorides that had precipitated during etching. The etched samples were sieved to >150 μm, to remove any partially dissolved feldspar grains. Aliquots for ESR measurement 20 comprised 60 mg of quartz loaded into glass tubes with interior and exterior diameters of 2 and 3 mm respectively.

## 4.2 Environmental dose rate determination

The grain size distribution of quartz and feldspar minerals within the parent bedrock was estimated from thin section analysis using the software of Buscombe (2013). The environmental dose rate, $\dot{D}$, was calculated from the sample specific radioisotope concentrations using DRAC v.1.2 (Durcan et al., 2015), the conversion factors of Guérin et al. (2011), the alpha grain size 25 attenuation factors of Bell (1980) and the beta grain size attenuation factors of Guérin et al. (2012). Because the bedrock samples have only been at the surface for a short period of time, no cosmic dose rate was included in the calculation. The water content was estimated at $2 \pm 2\%$. For the quartz extract, an etch depth of 10 μm was assumed and the alpha dose rate adjusted following Bell (1980); an a-value of $0.040 \pm 0.005$ was used after Rees-Jones (1995) for any residual alpha dose. No internal dose rate was included. In contrast, the feldspar fraction was not etched, and an a-value of $0.15 \pm 0.05$ was used after Balescu 30 and Lamothe (1994). An internal K-content of $12.5 \pm 5.0$ % was assumed following Huntley and Baril (1997). The calculated environmental dose rates are summarized in Table 1 and full calculation details are given in the Supplementary Material.



## 4.3 Electron Spin Resonance

Electron Spin Resonance measurements were done at the Leibniz Institute for Applied Geophysics in Hannover, Germany. Measurements were made on a JEOL JES-FA100 spectrometer using 2.0 mW microwave power, 0.1 mT modulation width, a 333.5 ± 15 mT magnetic field, 0.1 s time constant and 60 s scan which was averaged over 3 scans. All spectra were measured
3 times following sample turning by 60° to avoid any anisotropic effects. Measurements were made at -150 °C. The instrumentation detailed in Oppermann and Tsukamoto (2015) was used to facilitate X-ray irradiation and sample preheating, which is described below. The Ti and Al-centre peaks were fitted using V3.3.35 of the JEOL ESR data processing software, and were normalized relative to the intensity of the 6[th] hyperfine line of $Mn^{2+}$ from the internal MgO standard, doped with MnO. All subsequent data fitting was done using MATLAB.

## 10   4.3.1 Measurement protocol optimization

Tsukamoto et al. (2015; 2018) recently showed that it is necessary to preheat ESR samples that are measured in a single aliquot protocol to avoid any signal contribution from trapped charge that is unstable over laboratory timescales. Within this study, a series of tests were done to select the most appropriate preheat temperature and duration. The signal intensities of five aliquots of samples KRG16-06 and KRG16-104 were measured following different preheat treatments (i.e. one aliquot per temperature;
Fig. 2a). Aliquots of KRG16-06 were preheated for two minutes at temperatures of between 160 °C and 240 °C, whereas aliquots of sample KRG16-104 were preheated for four minutes at temperatures of between 120 °C and 200 °C. The signal intensity of a further aliquot of each sample was measured without laboratory preheating. In addition to measuring the ESR signal intensity, the equivalent doses of the Ti and Al-centres of KRG16-06 for each preheat temperature, were measured in a single aliquot method (Tsukamoto et al., 2015; Fig. 2b). The single aliquot protocol comprised measurement of the natural
signal, measurement of a single additive dose, annealing at 420 °C for two minutes and measurement following zero dose. All irradiations were given using an X-ray source with a dose rate of ~0.3 Gy s[-1] (Tsukamoto et al., 2018); aliquots were manually turned once during irradiation to ensure that even dosing was achieved.

## 25   4.3.2 Measurement of the trapped charge concentration

The trapped charge population of the different samples was measured using a single aliquot approach. This comprised measurement of the natural signal, a zero-point measurement following annealing of the aliquot at 380 °C for four minutes, and measurement of two or three regenerative doses points. The natural signal was then interpolated onto the dose response curve to determine the equivalent dose; all equivalent dose values were calculated using a linear fit. To confirm that the
measurement protocol was appropriate, a dose-recovery experiment was done. Three aliquots of zero-age sample KRG16-112 were given an X-ray dose of 360 Gy, before measurement using the same protocol outlined above. Trapped-charge dating





systems usually experience signal saturation, therefore it is also necessary to constrain the form of ESR centre dose response. Using a new aliquot of each sample, dose response was measured using the same measurement protocol, but omitting the zero-point measurement step.

### 4.3.3 Measurement of trapped charge thermal decay

Thermal signal losses were measured using an isothermal decay experiment, whereby three aliquots of each sample were irradiated with an additive dose of 4.30 kGy. The aliquots were then preheated at 160 °C for four minutes prior to initial measurement, and were then measured following isothermal holding at between 130 °C and 180 °C for 4, 8, 16, 32, 64, 128 and 256 minutes. This experiment was also repeated on three fresh aliquots of sample KRG16-104 using a smaller dose of 2.15 kGy.

**4.4 OSL measurements**

OSL measurements of all samples followed the approach of King et al. (2016b,c). Luminescence measurements were made at the University of Bern using a single aliquot regenerative dose multiple-elevated-temperature (MET) infra-red stimulated luminescence (IRSL) measurement protocol (Li and Li, 2011) comprising a preheat at 250 °C for 60 seconds, followed by four IRSL measurements at 50, 100, 150 and 225 °C each of 100 s duration. A test dose of 160 Gy was used, which is c. 30% of

the $IRSL_{50}$ signal equivalent dose value of samples KRG16-06, KRG16-101 and KRG16-104. Each measurement cycle was followed by a high temperature optical wash at 290 °C for 60 s. Regenerative doses up to ~4.50 kGy were given to three small (2 mm diameter) aliquots of each sample using two different Risø TL-DA-20 luminescence readers with dose rates ranging from 0.06 to 0.10 Gy s$^{-1}$ dependent on instrument (dose rates are provided for each measurement in the Supplementary Materials). Luminescence signals were detected in the blue part of the visible spectrum using a BG39 and BG3 or Corning 7-

59 filter combination. The suitability of the selected measurement protocol was confirmed using a dose recovery test.

Rates of athermal and thermal charge detrapping were also measured using a single aliquot regenerative dose method on the same aliquots used to measure the luminescence dose response curve. Athermal detrapping rates were quantified by measuring the luminescence response to a fixed dose following different delay periods. Aliquots were preheated prior to storage following

Auclair et al. (2003) and maximum fading delays were 122 days. Rates of thermal charge detrapping were measured using isothermal holding experiments. The aliquots were given a dose of 50 Gy, and held at temperatures ranging from 170 to 350 °C for delay times of 0 to 10,240 s prior to measurement.





## 5. Results

### 5.1 Electron Spin Resonance

The signal intensity experiment indicates a plateau for the Ti-centre of sample KRG16-104 up until 160 °C (Fig. 2a). In contrast, the Al-centre for this sample and the Ti-centre of sample KRG16-06 reduce in intensity by ~5% between room

temperature and 160 °C whilst the Al-centre of sample KRG16-06 is depleted further, by ~10%. The signal intensity data for KRG16-06 are relatively noisy (non-monotonic signal decay with increasing preheat temperature) in comparison to KRG16-104. However, in spite of this, within the preheat-plateau experiment (Fig. 2b), a plateau in $D_e$ values between 160 and 220 °C is recorded for this sample following preheating for two minutes.

Preheating for short durations resulted in the heater unit overshooting the target temperature and poor thermal reproducibility. For this reason, a longer duration preheat at 160 °C for four minutes was selected for all measurements. This selected protocol is further validated by the successful recovery of a 360 Gy dose from naturally zero-age sample KRG16-112 for both the Al and Ti-centres, which yield recovered to given dose ratios of $0.83 \pm 0.20$ and $1.01 \pm 0.06$ respectively (n=3).

Measurements of the trapped charge population of the Al and Ti-centres were similar between aliquots resulting in 1σ uncertainties of ~20%. Equivalent dose values for the Al- and Ti-centres were within uncertainty for all samples, and ages ranged from $247 \pm 54$ ka for sample KRG16-05 to $37 \pm 4$ ka for sample KRG16-104 (Table 1). Samples KRG16-111 and KRG16-112 from the high-temperature tunnel yielded zero age; consequently, full dose response and isothermal decay was not measured for sample KRG16-112. Whereas it was possible to saturate the Ti-centre of all samples with the maximum

given dose of 19 kGy, the Al-centre continued to grow linearly throughout measurement for all samples (Fig. 3a,c). Continued growth of the Al-centre has been reported previously and has been accommodated through fitting dose response with an exponential plus linear function (e.g. Duval, 2012). In contrast, for the KRG samples, the Al-centre is best described using a linear regression (Fig. 3a). In contrast, the Ti-centre of all samples showed a reduction in signal intensity at high doses (i.e. >10 kGy), which has also been reported previously (e.g. Duval and Guilarte, 2015) and has been attributed to changing electron

capture probabilities (Woda and Wagner, 2007). To characterise the maximum possible trapped-charge population we excluded data points where the ESR signal intensity started to reduce (white data points in Fig. 3c) and fitted the remaining data with a single saturating exponential function (e.g. Grün and Rhodes, 1991) of the form:

$$\tilde{n} \approx \frac{I}{I_{sat}} = (1 - e^{-\frac{D+D_e}{D_0}}) \qquad (11)$$


where $I$ is the natural ESR signal intensity, $I_{sat}$ is the saturation intensity of the ESR signal and $D$ is the given dose (Gy). Because the Ti-centre experiences saturation, the equivalent dose value, $D_e$, can also be expressed as a saturation ratio, i.e.





$\tilde{n} = (n/N)$. As only a single aliquot of each sample was dosed until saturation, $\tilde{n}$ values for the Ti-centre were calculated from interpolation of the average $D_e$ value (n=3) onto the single dose response curve, thus $\tilde{n}$ values for the Ti-centre are derived from multiple aliquots.

Toyoda and Ikeya (1991) suggested that the thermal decay of the *E'*, Al and Ti-centres follows second order kinetics, however it was not possible to fit our data using either a first or second order kinetic model (Supplementary Material). Instead the isothermal decay data were fitted using a multiple first order kinetic model (Lambert et al., In Review; Table 1). Whilst the actual physical meaning of a Gaussian distribution of energies requires further investigation within the context of ESR defects, and it is unlikely that both the Al and Ti-centres follow exactly the same process of thermal decay, preliminary fits to the data

using this model are promising (Fig. 3b,d). Values of $\mu(E_t)$ ranged from 1.3 – 1.9 eV between samples and centres (Table 1).

## 5.2 Optically Stimulated Luminescence

For all of the samples, the measured luminescence signals fulfilled the acceptance criteria (see Supplementary Material for further details). The $IRSL_{50}$ signals of all samples exhibited very high rates of fading with $g_{2days}$ values ranging from 6-11 %/decade, whereas for post-IR IRSL measurements at 225 °C, fading rates were 2-4 %/decade. The model introduced by Huntley (2006) was used to fit the athermal detrapping data to determine $\rho'$. Using this model to fading correct the trapped charge concentrations following Kars et al. (2008) indicates that the $IRSL_{50}$ signals of samples KRG16-06 and KRG16-101,

and all signals for sample KRG16-05 are saturated (see Supplementary Material). All other signals can be used to determine rock-cooling histories. The luminescence dose response data of all of the samples and signals were fitted with a single saturating exponential fit to determine the characteristic dose of saturation, $D_0$ and the concentration of trapped charge, $\tilde{n}$. Although for some samples a general order kinetic model (GOK) fit would result in lower deviation from the measured values, GOK fits have been shown to overestimate sample athermal field saturation values (King et al., 2018), which must be done

accurately to evaluate if a sample contains thermal information (cf. Valla et al., 2016). Finally, the isothermal decay data were fitted using the band-tail states model (Poolton et al., 2009; Li and Li, 2013; Eq. 6 and 7) to determine $E_t$, $E_u$ and $s$. Values of $E_t$ ranged from 1.2 – 1.5 eV between samples and signals (Table 2).

## 5.3 Inversion of ESR and OSL data for cooling histories

In order to invert the data into cooling histories, we used the same approach outlined in section 3. We computed OSL and ESR dose response curves from 10,000 randomly generated t-T paths, which were constrained to cool monotonically between 200 °C and 15 ± 5 °C, over 2 Ma. Initially the Al-centre, Ti-centre and OSL centres were inverted separately, before being inverted





together. The results for all samples with the exception of naturally zero-age samples KRG16-111 and KRG16-112, are shown in Fig. 4.

## 6. Discussion

Trapped-charge thermochronometers offer benefits over other thermochronometry systems because of their low closure temperature and ability to yield precise cooling histories over Quaternary timescales (Herman and King, 2018). However, signal saturation has proven a significant barrier to the application of luminescence thermochronometry (cf. Valla et al., 2016). For ESR thermochronometry to offer a viable alternative it should exhibit later signal saturation but also similar thermal stability. The measurements presented here are promising because whilst the ages measured for the OSL and ESR systems are similar, the maximum possible ages that can be obtained from the ESR Ti-centre are more than four times greater than the maximum possible age that can be obtained from the OSL signals (Tables 1 and 2). Furthermore, the Al-centre of the KRG samples does not exhibit signal saturation up to 19 kGy, which was the maximum dose explored in this study (Fig. 3a). Although such linear dose response behaviour has, to our knowledge, not been reported previously and thus may be a property of these exceptionally young quartz minerals, it is an exciting observation that warrants further study through the investigation of further quartz samples.

Samples KRG16-111 and KRG16-112 from the high temperature tunnel yielded zero, or near-zero ages for both ESR centres and the IRSL signals investigated (Tables 1 and 2). These samples provide an important local control on the thermal stability of these trapped-charge systems, demonstrating that all charge is evicted from the centres at sufficiently high temperatures. For the remaining samples, the ages obtained from the two ESR centres are within uncertainties, indicating that they may have similar thermal stability. For the OSL data, some variance in age is recorded between the different signals (Table 2); all signals of sample KRG16-05 are in field saturation, and thus only a minimum sample age of ~180 ka can be calculated (Table 2). The $IRSL_{50}$ signals of samples KRG16-06 and KRG16-101 are also in field saturation, yielding the highest apparent ages for these samples, and are not considered further. For samples KRG16-06 and KRG16-101, the remaining IRSL signals show a general reduction in age with increasing stimulation temperature, possibly indicating that the ages have been overcorrected for anomalous fading using the Huntley (2006) model (cf. King et al., 2018). The OSL and ESR ages of samples KRG16-06 and KRG16-104 are similar, indicating that for these samples the ESR and OSL signals have similar thermal stability, and thus that ESR-thermochronometry would also be suitable for resolving late stage cooling histories. In contrast, sample KRG16-101 yields OSL ages twice as large as the ESR ages, which could be indicative of a difference in centre thermal stability.

To further evaluate the relative thermal stability of the ESR and OSL signals, the isothermal decay of the ESR and feldspar systems was simulated using the experimentally constrained kinetic parameters of the different samples, for isothermal





conditions of 20 °C assuming an initial trapped charge concentration of 1 and assuming no charge trapping (Fig. 5). Note that anomalous fading related signal loss has also been included for the OSL signals, as excluding this variable would result in erroneously high apparent signal stabilities. The ESR centres have similar thermal stability to the IRSL centres for all samples, with the exception of sample KRG16-101 (Figs. 5c) where the ESR centres are more thermally stable. The Ti-centre is more

thermally stable than the Al-centre for all samples, with the exception of sample KRG16-104 for the measurement in response to 2.15 kGy. This is consistent with the earlier work of Grün et al. (1999) who also observed that the Ti-centre is more thermally stable than the Al-centre, but contrasts with observations from Chinese loess (Tsukamoto et al., 2018). The contrasting behaviour between the two measurements of KRG16-104 in response to doses of 4.30 kGy and 2.15 kGy (Fig. 5d) reflects uncertainty in the derivation of ESR kinetic parameters, potentially related to inter-aliquot variability. Improved measurement

protocols and the development of automated instrumentation may alleviate these discrepancies through improving measurement reproducibility, however despite this, the thermal stability determined in both experiments is broadly similar (Fig. 5d). The general trend of ESR signals exhibiting similar thermal stability to IRSL signals indicates that ESR-thermochronometry will record changes in exhumation histories from a similar thermal range as OSL-thermochronometry, whilst benefitting from considerably later signal saturation and being unaffected by anomalous fading.

Inverting the Al and Ti-centres of all samples results in broadly similar time-temperature histories between centres, whist the cooling histories of different samples vary (Fig. 4). This is in agreement with the Al and Ti-centres' similar thermal stabilities

(Fig. 5) and measured ages (Table 1). The two different centres can also be effectively combined to produce a single cooling history (Fig. 4), which is similar to that inverted from the OSL data alone for samples KRG16-101 and KRG16-104 (Fig. 4). For sample KRG16-05, the saturated OSL signals result in a broad cooling history, whereas for sample KRG16-06 the OSL data yield more rapid cooling than the ESR data. The OSL and ESR data can also be inverted together. These data show that for samples beyond the range of OSL dating, ESR-thermochronometry will be able to provide cooling histories over a similar

thermal range, allowing late stage exhumation histories to be determined. The data inversions reveal that rates of rock cooling in the Hida range of the Japanese Alps are consistent with previous investigations (Ito et al., 2013; 2017; Spencer et al., 2019). Whereas sample KRG16-05 experienced almost no cooling over the past 2 Myr, cooling rates accelerated from ~100 °C/Myr (calculated from the U/Pb ages of Ito et al., 2013) to rates of >400°C/Myr over the past 100 ka for samples KRG16-06, KRG16-101 and KRG16-104.

**7. Conclusions and Outlook**

In this study, the potential of ESR thermochronometry for constraining rates of rock cooling has been explored for a suite of samples from the Hida range of the Japanese Alps. Through using the latest ESR measurement protocols (Tsukamoto et al.,





2015) and instrumentation (Oppermann and Tsukamoto, 2015) the dose response and thermal stability of both the Al and Ti-centres has been constrained. Whilst the Ti-centre can be described with a single saturating dose response curve, the Al-centre continues to grow linearly with laboratory irradiation. A multiple-first order model based on a distribution of trap-depths was successfully used to fit isothermal decay data (Lambert et al., In Review), which do not follow either simple first-order, or

second-order decay. Contrasting the thermal stability of the Al and Ti-centres with that of the luminescence centres of feldspar shows that the ESR of quartz has similar thermal stability. The Al and Ti-centres can be successfully inverted together for rock cooling for all of the samples investigated. It was also possible to invert the OSL and ESR data together for all samples analysed, providing further constraints on their thermal histories. Whereas OSL-thermochronometry of sample KRG16-06 can only recover a minimum cooling rate of ~160 °C/Myr, both ESR centres have the potential to recover cooling rates of as low

as 50-25 °C/Myr, illustrating the potential of ESR for resolving late-stage cooling histories.

**Acknowledgements**

GEK acknowledges financial support from a Mobility Grant from the University of Cologne that funded initial fieldwork and from Swiss National Science Foundation (SNSF) grant number PZ00P2_167960. SS and TT acknowledge support from the Grant-in-Aid for Scientific Research on Innovative Areas (KAKENHI No.

26109003) from the Ministry of Education, Culture, Sports, Science and Technology (MEXT). We thank the Chubu Regional Environment Office for permission to collect rocks in the Chubu Sangaku National Park. Sample collection was supported by the Kansai Electric Power Co., Inc., Y. Hino (KANSO Co., Ltd.), Dr. T. Komatsu (JAEA), S. Terusawa (OYO Co., Ltd.), S. Fukuda, T. Arai (Kyoto Univ.), and staff of the Azohara lodge. The Herbette Foundation funded ST to stay at the University of Lausanne during the study. Benny Guralnik is thanked

for commenting on an earlier version of this manuscript.

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





**Table 1: ESR centre kinetic parameters, $D_e$ values and ages. Maximum ages are calculated from 2*$D_0$. Full details of environmental dose rate derivation are given in the Supplementary Material.**

| Ti-Centre | $\dot{D}$(Gy ka$^{-1}$) | $D_0$ (Gy) | $\mu(E_t)$ (eV) | $\log_{10}(s)$ (s$^{-1}$) | $\sigma(E_t)$ (eV) | $\tilde{n}$ | $D_e$ (Gy) | Age (ka) | Maximum Age (Ma) |
|---|---|---|---|---|---|---|---|---|---|
| KRG16-05 | 6.37 ± 0.38 | 2,555 ± 438 | 1.44 ± 0.10 | 12.72 ± 1.11 | 0.09 ± 0.01 | 0.50 ± 0.02 | 1,859 ± 81 | 291 ± 13 | 0.82 |
| KRG16-06 | 3.60 ± 0.13 | 3,182 ± 607 | 1.79 ± 0.10 | 17.16 ± 1.15 | 0.12 ± 0.01 | 0.20 ± 0.02 | 275 ± 24 | 76 ± 7 | 1.77 |
| KRG16-101 | 3.97 ± 0.33 | 2,495 ± 249 | 1.89 ± 0.15 | 17.88 ± 1.74 | 0.13 ± 0.01 | 0.10 ± 0.00 | 145 ± 6 | 37 ± 2 | 1.29 |
| KRG16-104 | 4.42 ± 0.23 | 2,804 ± 302 | 1.70 ± 0.15 | 15.42 ± 1.64 | 0.10 ± 0.01 | 0.16 ± 0.01 | 334 ± 15 | 76 ± 4 | 1.34 |
| KRG16-111 | 4.54 ± 0.42 | 2,915 ± 172 | 1.69 ± 0.07 | 15.89 ± 0.78 | 0.11 ± 0.01 | 0.00 ±0.00 | - | 4.76 ± 11.3* | 1.28 |
| **Al-Centre** | | | | | | | | | |
| KRG16-05 | 6.37 ± 0.38 | - | 1.27 ± 0.05 | 10.82 ± 0.58 | 0.09 ± 0.00 | - | 1,115 ± 56 | 175 ± 9 | |
| KRG16-06 | 3.60 ± 0.13 | - | 1.66 ± 0.06 | 15.95 ± 0.64 | 0.11 ± 0.00 | - | 267 ± 50 | 74 ± 14 | |
| KRG16-101 | 3.97 ± 0.33 | - | 1.90 ± 0.08 | 18.27 ± 0.93 | 0.10 ± 0.01 | - | 141 ± 11 | 36 ± 3 | |
| KRG16-104 | 4.42 ± 0.23 | - | 1.62 ± 0.13 | 14.60 ± 1.49 | 0.10 ± 0.01 | - | 307 ± 18 | 69 ± 4 | |
| KRG16-111 | 4.54 ± 0.42 | - | 1.58 ± 0.08 | 14.97 ±0 .88 | 0.10 ± 0.01 | - | - | 168 ± 251* | |

*Ages calculated from single aliquot additive dose response curve.



**Table 2: Summary of sample luminescence kinetic parameters. Full details of environmental dose rate derivation are given in the Supplementary Material. Ages in italics are saturated. Maximum ages are calculated from 2\*$D_0$.**

| Sample | Signal | $\dot{D}$ (Gy ka⁻¹) | $D_0$ (Gy) | $E_t$ (eV) | $\log_{10}(s)$ (s⁻¹) | $E_u$ (eV) | $\log_{10}(\rho')$ | $\tilde{n}$ | $\tilde{n}_{SS}$ | Age (ka) | Max. Age (ka) |
|---|---|---|---|---|---|---|---|---|---|---|---|
| KRG05 | IRSL50 | 8.57 ± 1.17 | 848 ± 29 | 1.33 ± 0.02 | 9.31 ± 0.21 | 0.07 ± 0.01 | -5.27 ± 0.08 | 0.25 ± 0.06 | 0.23 ± 0.07 | $404.62^{-93.11}_{251.75}$ | 184 |
|  | IRSL100 |  | 817 ± 20 | 1.38 ± 0.03 | 9.09 ± 0.24 | 0.07 ± 0.01 | -5.45 ± 0.06 | 0.36 ± 0.07 | 0.38 ± 0.05 | $243.05^{106.60}_{116.42}$ | 182 |
|  | IRSL150 |  | 837 ± 21 | 1.32 ± 0.04 | 8.02 ± 0.34 | 0.09 ± 0.01 | -5.73 ± 0.10 | 0.55 ± 0.06 | 0.60 ± 0.08 | $236.51^{326.66}_{70.10}$ | 191 |
|  | IRSL225 |  | 701 ± 21 | 1.34 ± 0.05 | 7.53 ± 0.43 | 0.13 ± 0.01 | -5.98 ± 0.06 | 0.65 ± 0.03 | 0.76 ± 0.03 | $157.96^{26.03}_{19.67}$ | 162 |
| KRG06 | IRSL50 | 7.10 ± 0.47 | 829 ± 39 | 1.36 ± 0.03 | 9.57 ± 0.25 | 0.06 ± 0.01 | -5.04 ± 0.04 | 0.06 ± 0.01 | 0.08 ± 0.02 | $121.37^{28.00}_{21.99}$ | 206 |
|  | IRSL100 |  | 1002 ± 39 | 1.41 ± 0.03 | 9.41 ± 0.28 | 0.07 ± 0.01 | -5.18 ± 0.05 | 0.07 ± 0.01 | 0.16 ± 0.03 | $67.40^{9.96}_{9.24}$ | 258 |
|  | IRSL150 |  | 980 ± 36 | 1.35 ± 0.04 | 8.28 ± 0.37 | 0.08 ± 0.01 | -5.45 ± 0.05 | 0.11 ± 0.01 | 0.38 ± 0.04 | $44.75^{4.36}_{4.22}$ | 265 |
|  | IRSL225 |  | 791 ± 32 | 1.41 ± 0.06 | 8.12 ± 0.49 | 0.13 ± 0.01 | -5.54 ± 0.04 | 0.18 ± 0.01 | 0.46 ± 0.04 | $52.31^{4.37}_{4.20}$ | 216 |
| KRG101 | IRSL50 | 6.57 ± 1.43 | 910 ± 46 | 1.52 ± 0.03 | 11.06 ± 0.12 | 0.07 ± 0.00 | -5.05 ± 0.04 | 0.08 ± 0.00 | 0.09 ± 0.02 | $217.99^{32.72}_{25.78}$ | 244 |
|  | IRSL100 |  | 1029 ± 44 | 1.39 ± 0.03 | 9.29 ± 0.27 | 0.08 ± 0.01 | -5.33 ± 0.03 | 0.11 ± 0.00 | 0.27 ± 0.03 | $71.51^{3.98}_{3.88}$ | 295 |
|  | IRSL150 |  | 1105 ± 45 | 1.40 ± 0.03 | 8.77 ± 0.25 | 0.09 ± 0.01 | -5.49 ± 0.05 | 0.13 ± 0.00 | 0.41 ± 0.04 | $61.60^{1.92}_{1.90}$ | 324 |
|  | IRSL225 |  | 892 ± 37 | 1.29 ± 0.04 | 7.14 ± 0.36 | 0.13 ± 0.01 | -5.77 ± 0.07 | 0.20 ± 0.00 | 0.63 ± 0.05 | $50.73^{1.37}_{1.36}$ | 267 |
| KRG104 | IRSL50 | 6.20 ± 0.85 | 784 ± 34 | 1.33 ± 0.03 | 9.14 ± 0.20 | 0.08 ± 0.01 | -5.19 ± 0.01 | 0.09 ± 0.01 | 0.17 ± 0.01 | $80.15^{15.72}_{13.84}$ | 231 |
|  | IRSL100 |  | 709 ± 29 | 1.37 ± 0.02 | 8.88 ± 0.21 | 0.09 ± 0.01 | -5.45 ± 0.03 | 0.16 ± 0.04 | 0.38 ± 0.02 | $57.57^{22.00}_{18.31}$ | 219 |
|  | IRSL150 |  | 777 ± 31 | 1.41 ± 0.03 | 8.93 ± 0.27 | 0.09 ± 0.01 | -5.57 ± 0.04 | 0.18 ± 0.04 | 0.48 ± 0.04 | $56.76^{16.97}_{14.88}$ | 243 |
|  | IRSL225 |  | 709 ± 31 | 1.39 ± 0.04 | 8.12 ± 0.34 | 0.13 ± 0.01 | -5.65 ± 0.07 | 0.24 ± 0.05 | 0.55 ± 0.05 | $63.32^{17.81}_{15.35}$ | 223 |
| KRG111 | IRSL50 | 7.03 ± 1.58 | 615 ± 32 | 1.38 ± 0.02 | 9.47 ± 0.19 | 0.08 ± 0.00 | -5.29 ± 0.02 | 0.00 ± 0.00 | 0.25 ± 0.02 | $0.64^{0.06}_{0.06}$ | 164 |
|  | IRSL100 |  | 871 ± 43 | 1.38 ± 0.02 | 8.12 ± 0.33 | 0.09 ± 0.01 | -5.63 ± 0.14 | 0.00 ± 0.00 | 0.53 ± 0.11 | $0.99^{0.04}_{0.04}$ | 241 |
|  | IRSL150 |  | 932 ± 39 | 1.40 ± 0.04 | 7.85 ± 0.56 | 0.09 ± 0.01 | -5.81 ± 0.22 | 0.01 ± 0.00 | 0.66 ± 0.16 | $1.16^{0.18}_{0.18}$ | 261 |
|  | IRSL225 |  | 748 ± 33 | 1.34 ± 0.05 | 6.46 ± 0.52 | 0.12 ± 0.01 | -5.86 ± 0.17 | 0.01 ± 0.00 | 0.69 ± 0.11 | $1.39^{0.32}_{0.32}$ | 210 |
| KRG112 | IRSL50 | 7.06 ± 1.54 | 572 ± 31 | 1.35 ± 0.02 | 9.90 ± 0.15 | 0.07 ± 0.00 | -5.32 ± 0.01 | 0.00 ± 0.00 | 0.28 ± 0.01 | $0.27^{0.02}_{0.02}$ | 151 |
|  | IRSL100 |  | 807 ± 37 | 1.34 ± 0.02 | 9.34 ± 0.18 | 0.08 ± 0.00 | -5.62 ± 0.10 | 0.00 ± 0.00 | 0.52 ± 0.08 | $0.32^{0.08}_{0.08}$ | 221 |
|  | IRSL150 |  | 847 ± 32 | 1.36 ± 0.03 | 9.12 ± 0.27 | 0.10 ± 0.00 | -5.92 ± 0.32 | 0.00 ± 0.00 | 0.72 ± 0.21 | $0.36^{0.12}_{0.12}$ | 236 |
|  | IRSL225 |  | 768 ± 35 | 1.27 ± 0.04 | 7.78 ± 0.33 | 0.12 ± 0.01 | -6.04 ± 0.24 | 0.00 ± 0.00 | 0.78 ± 0.13 | $0.44^{0.08}_{0.08}$ | 215 |





**Figure 1: Synthetic inversions of ESR and OSL data for monotonic cooling of 100 °C/Myr, 75 °C/Myr, 50 °C/Myr, 25 °C/Myr and no cooling. (a) Cooling histories and (b) forward modelled Ti (primary y-axis) and Al (secondary y-axis) centre signal accumulation, (c) OSL centre signal accumulation. ESR and OSL signals after 2 Myr were then inverted to derive cooling histories for the Al and Ti and OSL centres, as well as for the Al and Ti-centres combined for the different cooling scenarios. The original cooling history from (a) is shown as a white dashed arrow in each of the cooling histories, whilst the 1 and 2σ and median cooling histories are shown in green, black and red respectively.**




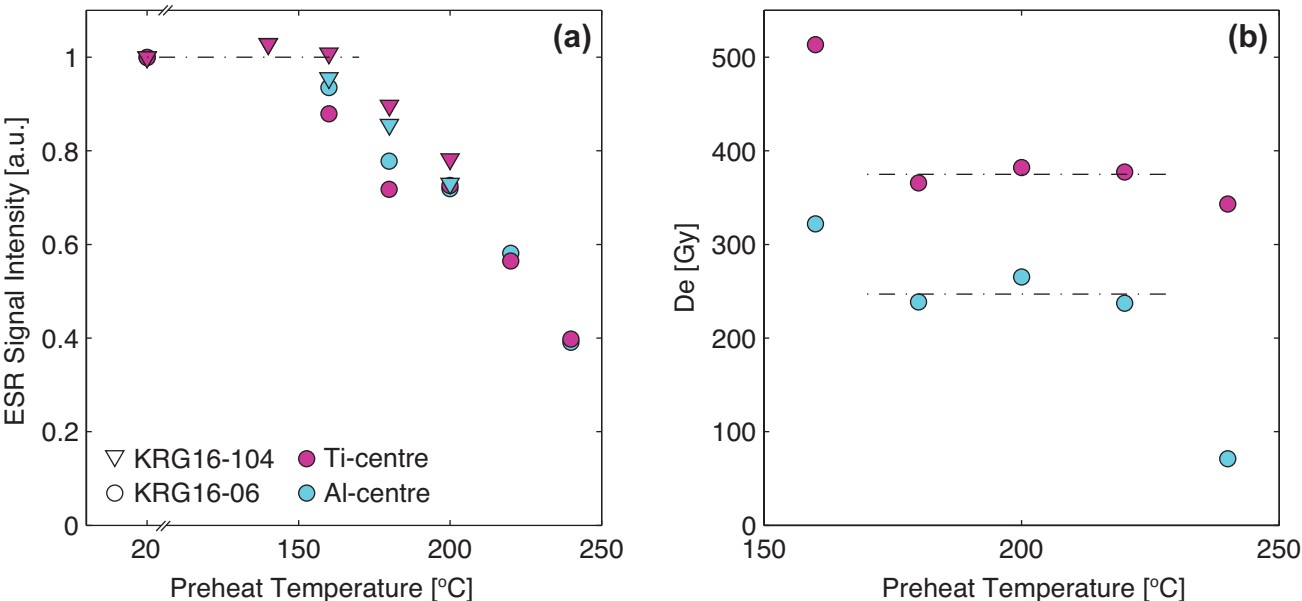

**Figure 2: (a) Changing ESR signal intensity with increasing preheat temperature for KRG16-06 (two minute preheats) and KRG16-104 (four minute preheats). Signal intensities are normalised relative to measurements made following no preheating (shown at 20 °C). (b) Preheat plateau data for KRG16-06 based on measurement of a single aliquot at each temperature (two minute preheats).**



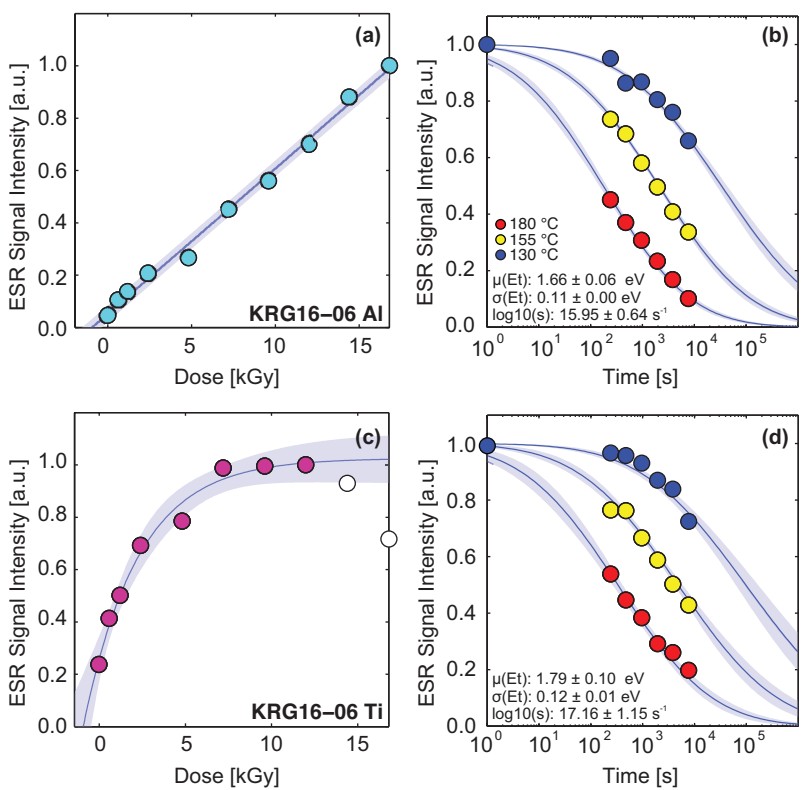

**Figure 3: ESR dose response and isothermal decay for the Al (a,b) and Ti-centres (c,d) of sample KRG16-06. Whereas the Al-centre (a) experiences linear signal accumulation, the Ti-centre (b) follows exponential growth before the signal intensity starts to reduce (white data points). This reduction in signal intensity is thought to represent radiation dose quenching of the ESR signal (cf. Woda and Wagner, 2007; Tissoux et al., 2007; Duval and Guilarte, 2015) and these data were excluded before fitting. The isothermal decay of the Al (a) and Ti-centres (b) is fitted with a density of states model assuming a Gaussian distribution around trap depth (Lambert et al., In Review).**



**Figure 4: Probability density functions of cooling histories inverted from the ESR and OSL data of samples KRG16-05, KRG16-06, KRG16-101 and KRG16-104. The different rows show inversion of the Al-centre, the Ti-centre, the Al and Ti-centres together, all four OSL signals (i.e. $IRSL_{50}$, $IRSL_{100}$, $IRSL_{150}$, $IRSL_{225}$) and finally the Al and Ti-centres, and the OSL centres together. Time-temperature histories were generated over 2 Myr with random monotonic cooling from 200 °C to 15 ± 5 °C. All probability density functions are scaled relative to 1. Model residuals for the inversion of all signals together are shown in the Supplementary Material.**



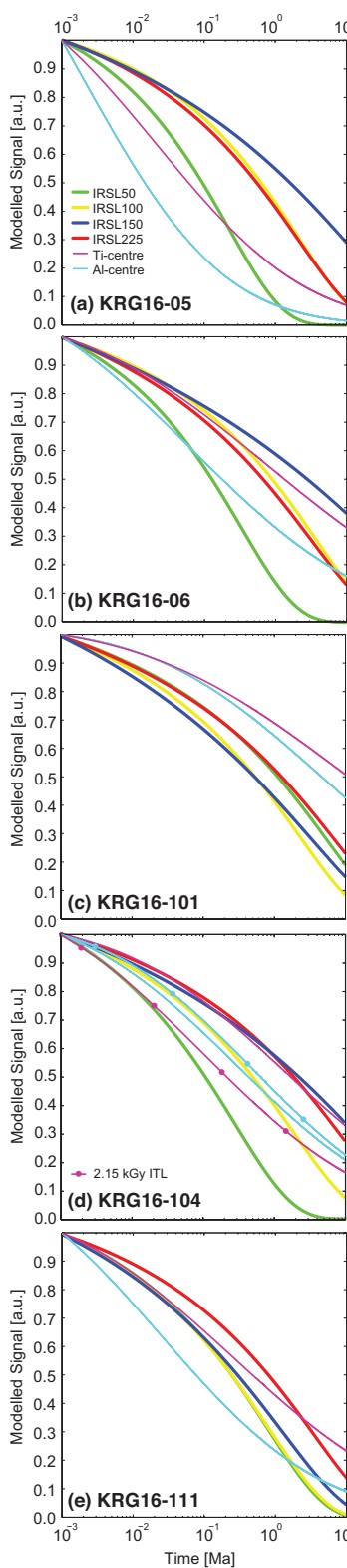

**Figure 5: Thermal stability of ESR signals in comparison to IRSL signals. The isothermal decay of (a) KRG16-05, (b) KRG16-06, (c) KRG16-101, (d) KRG16-104 and (e) KRG16-111 were modelled using the kinetic parameters listed in Tables 1 and 2 under isothermal conditions of 20 °C. Anomalous fading signal loss has been included in modelling of the IRSL data.**