# Peer review of "ESR-thermochronometry of the Hida range of the Japanese Alps: Validation and future potential"

_Geochronology, 2019_

## Referee Comment (RC1) · Nathan Brown (Referee) · 29 Jul 2019

This study is a thorough and compelling application of ESR to low-temperature thermochronology to the Japanese Alps. The authors have put forward a tremendous effort to present a detailed comparison of multi-elevated-temperature IRSL results from feldspar against Al and Ti ESR centers within quartz. The results are encouraging and provide the reader with a broad overview of why such an advance is worthwhile (similar stability to OSL signals but applicable for slower cooling rates), while still carefully outlining methodology limitations (e.g., lack of automated measurement capabilities and uncertainties associated with reaction kinetics). Below, I offer several questions I had while reading along with a few suggestions or concerns.

[Figure]

Main text:

p.2,l.20: The averaging time(s) would be helpful for these rates.

p.2,l.30: Slightly unclear what 'paired' means in this context.

p.3,l.4: Here you could also mention that signal intensity is a persistent limitation for quartz OSL thermochronometry (unlike for ESR, apparently).

p.3,Eq.10ff.: The negative sign before the activation energy is difficult to see with the current typesetting.

p.3,l.30: "...a model that assumes a Gaussian distribution of activation energies, $E_a$ around the mean trap depth, $\mu(E_t)$(eV)." It would be good to also mention the meaning of $\sigma(E_t)$ here.

Besides the Lambert study in review, is there any precedent in ESR literature for treating the activation energy in this way? The reason for adopting this approach probably deserves either an available citation or further justification in the main text, even if only a sentence or two. The full explanation within the Supplementary Materials is excellent, but a quick note here would be good.

p.4,ll.8-9: Is this owing to the long irradiation times with common beta sources?

p.5,l.23: It seems misleading to label alpha as a 'constant' when it has a known functional form (e.g., Chen and McKeever, 1997, pp. 60-66) that depends upon the trap depth (which is often allowed to vary in studies such as this).

p.5,l.27: I believe that 'charge' encompasses electrons and electron holes.

p.6,ll.20-22: It seems important to qualify here that this result hinges on the assumption of a correct kinetic expression; this statement should not be misunderstood to mean that the authors have (at this stage in the manuscript) successfully recovered age information from slowly cooling samples, but that, to the degree that the kinetic expressions are accurate, slow cooling histories should be within resolution.

p.9,l.22: Strictly, you have quantified the room temperature detrapping. Presumably there is little thermal detrapping involved, but might be worth mentioning briefly to avoid confusion with LNT fading measurements.

p.11,l.19: That IRSL_50 signals are saturated and higher temperature signals are unsaturated in the same sample seems to me an inexplicable result. Can you comment on why this might be observed?

p.11,l.24: I could not find the King et al. (2018) citation within the references. 'Athermal field saturation values' seems to be an inappropriate concept. Even for traps which are considered stable over burial timescales (e.g., qz fast component), we still discuss 'trap lifetimes.' The same practice should apply for thermal detrapping that happens within feldspars at Earth's surface. Field saturation should therefore be understood to reflect athermal and thermal loss processes, even if athermal loss is expected to be dominant at lower temperatures.

p.11,l.32: Given that samples were taken from a transect that spans 1.2 km of elevation gain, shouldn't we expect more (and systematic) temperature variation with elevation? Most adiabatic lapse rates result in temperature loss of just under 10C per km of gained elevation.

p.12,ll.27-30: Is such variation in thermal stability between ESR and IRSL populations expected from previous work? Also, is there a reason to use OSL and IRSL interchangeably? I find it a little confusing and would prefer simply referring to 'luminescence' signals or IRSL results.

p.13,l.6: Please also mention that Grun et al. (1999) extracted quartz from granite.

Supplementary Materials:

From a physical standpoint, I'm a little dubious about the prediction that GOK decay predicts dose-dependent decay in a non-saturating system. The formulation of second-order kinetics (Garlick and Gibson, 1948) was developed for a phosphor where retrapping was predicted to increase dramatically as traps filled, therefore slowing recombination in a way that increases with dose. If, however, there is no upper limit to available trapping sites, this limitation should disappear. Therefore, I am skeptical that the transition from Eq.S3 to Eq.S7, while mathematically sound, is physically sensible.

p.5,ll.4-8: Wow! This difference in stability between OSL and ESR centers between samples is a really intriguing result!

p.5,l.17: "For all samples, the BTS model predicts..." Are there not many kinetic assumptions built into this prediction, including the nature and shape of the band-tail? In other words, could a higher stability be predicted if the tail were assumed to be quadratic or if the tailing factor were higher? Or, are all of these values sufficiently quantified for these samples? Perhaps this is what you reference in the final sentences of this paragraph?

---

## Referee Comment (RC2) · Anonymous Referee #2 · 26 Sep 2019

Comments on King et al. submitted to Geochronology

This is the first paper reporting the thermochronology with both luminescence and ESR dating techniques. I strongly recommend this paper to be published after clarifying the points below.

(1) Last part of Chapter 3 and latter half of Fig.1: The authors once calculated the change with time of each signal intensity (Figs. 1b and c), then, using these results, they inverted to obtain the predicted cooling histories. Therefore, ideally, the red lines in latter half of Fig. 1 matches the white dashed lines, if I understand correctly. However, some of them are not. The authors should explain and discuss this point more clearly. It would partly because of the assumed initial condition, but there are cases that cannot be accepted, especially, slowest cooling rates for OSL centers. Probably,

[Figure]

the discussion should be as such, in a case that the predicted cooling history obtained from OSL centers does not match that of ESR centers, the latter should be adopted. Then, this shall be applied to actual cases, i.e., Fig. 4.

(2) Discussions for Fig. 4: Probably, for samples KRG16-101 and 104, the results for all signals seems consistent, however, for the other two samples, they look inconsistent. The authors may use the criterion in (1), or may abandon the modelling. There should be cases that the results from different signals are not consistent with each other, then modelling of cooling history cannot be made from the statistical point of view. Probably Eq 10 would be for this. What are the L values for these?

Detailed points

Page 2 line 6: "later" should be "at higher doses"

Page 3 Eqs.1 and 2: "$E_a - \mu(E_t)$" should be "$E_a$".

Page 5 eq. 5: The first term, "$E_a$" should be "$E_b$", second term, "$E_t - E_b$" should be "$E_b$".

Page 6 line 1: What is n_mod?

Page 6 lines 1-8: What is m? Probably number of traps.

Page 6 Eq. 9: Is this summation from 1 to m? If so, it is not clear.

Page 7 line 17: Correct the inequality sign.

Page 8 line 7: "fitted" is by the least square method? How Ti-Li and Ti-H centers ratio was assumed?

Page 10 line 3: What is "signal intensity experiment"?

Page 10 line 3: Is "plateau" preheat plateau?

Section 5.1: One example of observed ESR spectrum should be shown together with a fitted spectrum.

Page 10 line 17: Correct the values and/or sample number.

Page 10 line 19: KRG16-112 is not listed in the Table.

Page 10 lines 20-23: Section 4.3.1 probably says the authors adopted regenerative protocol, but the dose response in Figs. 3 are additive dose.

Page 13 line 1, Fig. 5: The signals seem to reduce too much. Please check the number in horizontal axis.

Page 13 line 26, "consistent": Please describe how consistent.

---

## Author Response (AR1)

Nathan Brown (Referee) nathan.brown@berkeley.edu

This study is a thorough and compelling application of ESR to low-temperature thermochronology to the Japanese Alps. The authors have put forward a tremendous effort to present a detailed comparison of multi-elevated-temperature IRSL results from feldspar against Al and Ti ESR centers within quartz. The results are encouraging and provide the reader with a broad overview of why such an advance is worthwhile (similar stability to OSL signals but applicable for slower cooling rates), while still carefully outlining methodology limitations (e.g., lack of automated measurement capabilities and uncertainties associated with reaction kinetics). Below, I offer several questions I had while reading along with a few suggestions or concerns.

Main text:
p.2,l.20: The averaging time(s) would be helpful for these rates.

*Unfortunately this information is not given in the original publication, so it is not possible to add it to the text. As the saturation limits of the sample are also not described in the original paper, it is not possible for us to calculate the validity of these cooling rates for a particular time period.*

p.2,l.30: Slightly unclear what 'paired' means in this context.

*Removed paired.*

p.3,l.4: Here you could also mention that signal intensity is a persistent limitation for quartz OSL thermochronometry (unlike for ESR, apparently).

*As we have only explored this sample set at this point in time, we are reluctant to make a general statement about the behaviour of quartz ESR-thermochron signals. For this reason we respectfully choose not to make this addition.*

p.3,Eq.10ff.: The negative sign before the activation energy is difficult to see with the current typesetting.

*We have added an additional space which makes the negative sign clearer.*

p.3,l.30: "...a model that assumes a Gaussian distribution of activation energies, $E_a$ around the mean trap depth, $\mu(E_t)(eV)$." It would be good to also mention the meaning of $\sigma(E_t)$ here. Besides the Lambert study in review, is there any precedent in ESR literature for treating the activation energy in this way? The reason for adopting this approach probably deserves either an available citation or further justification in the main text, even if only a sentence or two. The full explanation within the Supplementary Materials is excellent, but a quick note here would be good.

*We believe that it is the first time that ESR signals have been modelled in this way. We added the following to the main text:*

*An alternative approach could be to use a first or second order kinetic model as has been done previously (Toyoda and Ikeya, 1991; Ikeya, 1993; Grün et al., 1999) and we discuss our model selection more completely in the supplementary material.*

p.4,ll.8-9: Is this owing to the long irradiation times with common beta sources?

*Because of the comparatively large volume of sample measured in ESR dating, it is not possible to use a beta source for irradiation. Multiple aliquot methods relate to the physical distance between gamma sources and measurement facilities, which mean that it is more convenient to dose multiple aliquots prior to measurement which must be done elsewhere. We have amended the sentence to make this clearer:*

*The former approach has only recently been made practical, following the introduction of X-ray irradiation for regenerative dosing (Oppermann and Tsukamoto, 2015), as opposed to gamma irradiation which is often done at a laboratory separate to the measurement laboratory.*

p.5,l.23: It seems misleading to label alpha as a 'constant' when it has a known functional form (e.g., Chen and McKeever, 1997, pp. 60-66) that depends upon the trap depth (which is often allowed to vary in studies such as this).

*Amended to "alpha is a constant related to the Bohr radius of the electron trap".*

p.5,l.27: I believe that 'charge' encompasses electrons and electron holes.

*Agreed and amended.*

p.6,ll.20-22: It seems important to qualify here that this result hinges on the assumption of a correct kinetic expression; this statement should not be misunderstood to mean that the authors have (at this stage in the manuscript) successfully recovered age information from slowly cooling samples, but that, to the degree that the kinetic expressions are accurate, slow cooling histories should be within resolution.

*We think that this is implicit in the exercise performed here as it also holds for the OSL data described in the preceding lines, and prefer not to make any further qualification.*

p.9,l.22: Strictly, you have quantified the room temperature detrapping. Presumably there is little thermal detrapping involved, but might be worth mentioning briefly to avoid confusion with LNT fading measurements.

*We have inserted "at room temperature" to be explicit that these measurements were not made at LNT.*

p.11,l.19: That IRSL_50 signals are saturated and higher temperature signals are unsaturated in the same sample seems to me an inexplicable result. Can you comment on why this might be observed?

*This is not an uncommon observation for thermochronometry data (although much of these data remain unpublished at this time). It is simply because the IRSL50 signals exhibit much greater rates of fading than the higher temperature signals, meaning that they reach athermal steady state more rapidly. We have added this sentence for clarification:*

*"Saturation of the IRSL$_{50}$ signals relative to the higher temperature signals is a consequence of their relatively high rate of anomalous fading"*

p.11,l.24: I could not find the King et al. (2018) citation within the references. 'Athermal field saturation values' seems to be an inappropriate concept. Even for traps which are considered stable over burial timescales (e.g., qz fast component), we still discuss 'trap lifetimes.' The same practice should apply for thermal detrapping that happens within feldspars at Earth's surface. Field saturation should therefore be understood to reflect athermal and thermal loss processes, even if athermal loss is expected to be dominant at lower temperatures.

*We follow the nomenclature of Kars et al. (2008); Valla et al. (2016) – QG and King et al. (2016) – QG. It is not necessary to invoke any thermal loss to explain the trapped charge concentrations of these samples, hence we refer to the athermal field saturation values. These values were calculated assuming no thermal losses using equation 8 in King et al. (2016) which is equivalent to equation 15 of Li and Li (2008). If we recalculated the trapped-charge concentrations using the full differential equation, i.e. including both thermal and athermal losses, the values would not change significantly, reflecting low rates of thermal detrapping at the surface temperatures experienced in this region. We prefer not to amend the text which clearly states that these values are calculated on an athermal basis. The citation to King et al. (2018) has now been included in the references.*

p.11,l.32: Given that samples were taken from a transect that spans 1.2 km of elevation gain, shouldn't we expect more (and systematic) temperature variation with elevation? Most adiabatic lapse rates result in temperature loss of just under 10C per km of gained elevation.

*This is an interesting point that we have considered carefully. We were unable to find an average adiabatic lapse rate for the Japanese Alps in the literature, however as the climate here is humid, it is likely that the adiabatic lapse rate is closer to 5°C per km, rather than 10°C per km. As such, we think that our allowed uncertainty of ±5°C on our final temperature for the inversion is appropriate.*

p.12,ll.27-30: Is such variation in thermal stability between ESR and IRSL populations expected from previous work? Also, is there a reason to use OSL and IRSL interchangeably? I find it a little confusing and would prefer simply referring to 'luminescence' signals or IRSL results.

*We are unaware of previous work that has compared the ESR and luminescence signal stability of the same samples. This is something that we plan to pursue in future research. We found it encouraging that using the relative age differences between the luminescence and ESR signals, we could then predict what the relative difference in thermal stability should be, and that our experimental data was consistent with these predictions.*

*We follow Guralnik et al. (2015) – EPSL in using OSL to refer to generic optically stimulated luminescence signals, rather than being more specific regarding the type of stimulation (i.e. infrared stimulation).*

p.13,l.6: Please also mention that Grun et al. (1999) extracted quartz from granite.

*Amended.*

Supplementary Materials: From a physical standpoint, I'm a little dubious about the prediction that GOK decay predicts dose-dependent decay in a non-saturating system. The formulation of second order kinetics (Garlick and Gibson, 1948) was developed for a phosphor where retrapping was predicted to increase dramatically as traps filled, therefore slowing

recombination in a way that increases with dose. If, however, there is no upper limit to available trapping sites, this limitation should disappear. Therefore, I am skeptical that the transition from Eq.S3 to Eq.S7, while mathematically sound, is physically sensible.

*This is a good point but for completeness we feel that is important to include this equation and consideration in the supplementary materials.*

p.5,ll.4-8: Wow! This difference in stability between OSL and ESR centers between samples is a really intriguing result!

p.5,l.17: "For all samples, the BTS model predicts..." Are there not many kinetic assumptions built into this prediction, including the nature and shape of the band-tail? In other words, could a higher stability be predicted if the tail were assumed to be quadratic or if the tailing factor were higher? Or, are all of these values sufficiently quantified for these samples? Perhaps this is what you reference in the final sentences of this paragraph?

*Here when we refer to BTS or GAU a certain distribution of band-tails is implicit. I.e. for the BTS we assume an exponential distribution of band-tails below the conduction band, whereas for GAU we instead assume a distribution of activation energies around the trap depth. The absolute value of the distribution (i.e. band-tail width, or width of the Gauss distribution around the trap depth) is determined from fitting the isothermal holding data for that particular sample and system. The GAU model predicts higher thermal stability than the BTS model, because it assumes a different energy distribution.*

*We have added "sample-specific" to make it clearer that these values are calculated for the particular sample under investigation. The distributions assumed for the GAU and BTS models are described in equations 4 and 7 respectively.*

Anonymous Referee #2
This is the first paper reporting the thermochronology with both luminescence and ESR dating techniques. I strongly recommend this paper to be published after clarifying the points below.

(1) Last part of Chapter 3 and latter half of Fig.1: The authors once calculated the change with time of each signal intensity (Figs. 1b and c), then, using these results, they inverted to obtain the predicted cooling histories. Therefore, ideally, the red lines in latter half of Fig. 1 matches the white dashed lines, if I understand correctly. However, some of them are not. The authors should explain and discuss this point more clearly. It would partly because of the assumed initial condition, but there are cases that cannot be accepted, especially, slowest cooling rates for OSL centers. Probably, the discussion should be as such, in a case that the predicted cooling history obtained from OSL centers does not match that of ESR centers, the latter should be adopted. Then, this shall be applied to actual cases, i.e., Fig. 4.

*Thank you for raising this important point. The reviewer is correct that the exercise shown in Fig. 1 tests whether the prescribed white line cooling histories can be recovered by inverting the forward modelled data shown in Figs 1a-c. The mis-match between the red and white lines provides some indication of signal performance, however the red-line is the median model of the accepted cooling histories used to generate the probability density function shown and rather the comparison should be made relative to the white line and brightly shaded parts of the PDFs. We have amended the latter part of section 3.2 to make this clearer by adding two sentences:*

*The results of the forward modelling and the synthetic inversions for the ESR and OSL data are shown in Fig. 1. The OSL signals for all cooling histories reach saturation (Fig. 1c), and this is reflected in the failure of the OSL to recover any of the cooling histories when inverted. This is*

*apparent because the 1σ confidence intervals show a broad range, with the highest density of cooling histories concentrated at temperatures < 20 °C over the past 500 ka indicating that the luminescence signals are saturated (as shown in Fig. 1c). The minimum cooling rate that can be resolved using OSL for sample KRG16-06 is ~160 °C/Myr, calculated from 86% of the luminescence signal saturation level. Signal saturation is the key limitation that restricts the application of luminescence thermochronometry to regions undergoing rapid exhumation. In contrast, it is clear that the ESR data are able to resolve the 100 °C/Myr, 75 °C/Myr and 50 °C/Myr synthetic cooling histories, and cooling rates of 25 °C/Myr are distinct from isothermal holding at 0 °C over timescales of ~2 Ma. This is apparent because of the coincidence between the prescribed cooling histories (white lines) and the highest density of accepted cooling histories shown by the brightest colours in the probability density functions. These results are significant as they show that ESR-thermochronometry is applicable in a range of geological settings beyond the rapidly exhuming locations that luminescence-thermochronometry is currently restricted to.*

(2) Discussions for Fig. 4: Probably, for samples KRG16-101 and 104, the results for all signals seems consistent, however, for the other two samples, they look inconsistent. The authors may use the criterion in (1), or may abandon the modelling. There should be cases that the results from different signals are not consistent with each other, then modelling of cooling history cannot be made from the statistical point of view. Probably Eq 10 would be for this. What are the L values for these?

*The reviewer is correct that it is easier to combine some ESR/OSL signals than others. The reason is likely a combination of factors including natural sample variability, experimental data and/or numerical model limitations. In order to treat the data objectively, all numerical modelling was done under the same conditions i.e. with the same initial condition, over the same time-period with the same range of final temperatures and for the same number of iterations. As it was comparatively challenging to fit KRG16-06 fewer cooling histories were accepted after the values of L (Eqs. 9 and 10) were treated with the rejection algorithm i.e. L is contrasted with a random number between 0 and 1, if L is greater the cooling history is retained. The benefit of using this approach is that the full range of possible cooling histories are accepted. If the data could not be fitted, no cooling histories would be accepted, and this is something that we have observed multiple times for OSL data, but not for the samples presented here.*

Detailed points
Page 2 line 6: "later" should be "at higher doses"

*Amended.*

Page 3 Eqs.1 and 2: "Ea – μ(Et)" should be "Ea".

*Thank you for spotting this. We have amended the equation.*

Page 5 eq. 5: The first term, "Ea" should be "Eb", second term, "Et – Eb" should be "Eb".

*Thank you for spotting this. Ea should be Eb. Amended. However Et-Eb should not be Eb. Eb is the energy of the particular band-tail state, the total energy to escape the trap is Et-Eb i.e. the trap-depth minus the energy of the band-tail width.*

Page 6 line 1: What is n_mod?

*We have clarified the sentence so that it reads:*

*"For each t-T path we calculated a misfit between the final inverted trapped-charge population, $\tilde{n}_{mod}$, and our forward modelled values, $\tilde{n}_{fwd}$ (Wheelock et al., 2015),"*

*Thus n_mod is the final inverted trapped-charge population.*

Page 6 lines 1-8: What is m? Probably number of traps.

*Yes, this is defined on line 6 of the original submission "for m traps".*

Page 6 Eq. 9: Is this summation from 1 to m? If so, it is not clear.

*This is summed over m traps. We do not know how to make the nomenclature clearer.*

Page 7 line 17: Correct the inequality sign.

*Sorry, however we are unsure what the reviewer is referring to. The quartz extracts have a density >2.58 and <2.70 hence $2.58 > \rho < 2.70$ g cm$^{-3}$.*

Page 8 line 7: "fitted" is by the least square method? How Ti-Li and Ti-H centers ratio was assumed?

*Our measurements were carried out at -150 °C which meant that we were unable to differentiate between the Ti-Li and Ti-H centres. We have added a sentence stating this explicitly to the text.*

*"As our measurements were carried out at -150 °C, it was not possible to differentiate between the Ti-H and Ti-Li centres, and consequently they have been treated as a single centre."*

Page 10 line 3: What is "signal intensity experiment"?

*The signal intensity experiment is described in section 4.3.1 (page 9, line 14 of the original submission). It comprised measurement of how the signal intensity changed as a function of changing preheat treatment.*

Page 10 line 3: Is "plateau" preheat plateau?

*Yes, although not in the usual sense. It is a plateau in signal intensity with changing preheat temperature. We feel this is clear from the sentence "The signal intensity experiment indicates a plateau for the Ti-centre of sample KRG16-104 up until 160 °C"*

Section 5.1: One example of observed ESR spectrum should be shown together with a fitted spectrum.

*We have added a new figure 2 showing an ESR spectrum for sample KRG16-06 and how the Al and Ti-centres were fitted.*

Page 10 line 17: Correct the values and/or sample number.

*Thank you for spotting this. Text updated.*

Page 10 line 19: KRG16-112 is not listed in the Table.

*No. This sample was used only in the dose recovery test. Therefore, it is not possible to include its details in this table. We amended the text on page 10, line 19 of the original submission to read "full dose response and isothermal decay was not measured for sample KRG16-112 and it is not included in Table 1"*

Page 10 lines 20-23: Section 4.3.1 probably says the authors adopted regenerative protocol, but the dose response in Figs. 3 are additive dose.

*The reviewer is correct and we tried to be explicit about this in the text. We measured the De values of our samples using a regenerative protocol as detailed in section 4.3.1. However, for measurement of the dose response into saturation we used an additive dose protocol. We have clarified this in the final sentence of section 4.3.2:*

*"Using a new aliquot of each sample, dose response was measured using the same measurement protocol, but omitting the zero-point measurement step, i.e. in an additive dose response protocol."*

Page 13 line 1, Fig. 5: The signals seem to reduce too much. Please check the number in horizontal axis.

*We re-checked these calculations and the figures are correct. Both the band-tail states model for luminescence and the GAUSS model used here for the ESR samples predict lower thermal stability than a single first order kinetic model.*

Page 13 line 26, "consistent": Please describe how consistent

*Qualifying how consistent the data are is difficult within the remit of this study, and is the focus of currently ongoing work. Ito et al. (2013; 2017) and Spencer et al. (2019) reported extremely young U-Pb and Zircon (U-Th-He) ages for this site. The fact that the luminescence data are not saturated is consistent with this. We have tried to give further information, whilst avoiding a lengthy discussion on this topic, which we feel is outside of the remit of this study, by amending this sentence to:*

*"The data inversions reveal that rates of rock cooling in the Hida range of the Japanese Alps are consistent with previous investigations that indicate rapid rock cooling (Ito et al., 2013; 2017; Spencer et al., 2019)."*

**Associate Editor comments**

The authors' response to the reviewers' comments were mostly satisfactory and helped clarify the discussion. The one exception is one reviewer's complaint about the use of OSL as a generic name for all photon-induced luminescence. The authors' defended such use based on past practice. There may be past practice but I think it is a bad idea, and such practice should not be propagated. I was confused by the authors talking about OSL when they meant IRSL. I think the authors should use "luminescence" as a generic term, and use OSL meaning inducement by visible light and IRSL meaning inducement by infrared light.

*We tested changing OSL to luminescence, but think that this will cause confusion. Revising OSL to luminescence makes the description of the analytical method even more vague (optical or thermal stimulation?) and further makes the reference to previous work which has been titled as OSL-thermochronometry (Guralnik et al., 2015-EPSL; King et al., 2016-QG; 2016-*

*Science; Herman and King, 2018-Elements) more challenging. If we refer to multi-IRSL-thermochronometry then it sounds like a new approach, which it is not. Equally, if we state IRSL, rather than OSL, it implies IRSL$_{50}$ rather than post-IR IRSL, or a MET protocol. For these reasons we remain of the opinion that OSL is a more appropriate term within the current manuscript.*

Thermochronometry is a fairly specialized branch of trapped-charge dating, and I think the authors should not assume all their readers are such specialists. I think, in particular, they should describe briefly in non-mathematical terms the distinction between forward modeling and inverse modeling. I believe the former refers to modeling based on kinetic parameters and the latter to modeling based on dose response curves, but I am not sure. Explaining that will go along ways toward making the paper more accessible to non-specialists.

*Thank you for this important comment. We have modified section 3 to provide this clarification:*

*"It is thus necessary to verify that ESR-thermochronometry offers advantages over luminescence methods. To achieve this, a series of synthetic tests for known cooling histories were done using the kinetic parameters of sample KRG16-06 (Table 1). These tests first comprised running a forward model, which uses sample-specific kinetic parameters and a rate equation to describe signal growth. Through forward modelling, it is possible to predict the trapped-charge concentration for a particular cooling history. The second stage of the test comprised inverting the trapped-charge concentrations predicted by the forward model, using the same rate equation, to determine if it is possible to recover the cooling history used in the forward model prediction. Further details of the forward and inverse modelling are given below."*

I found the discussion on intensity preheat plateau and equivalent dose preheat plateau somewhat opaque. One of the reviewers was also confused by this. The authors should explain better the purpose of the intensity plateau test.

*I have included a citation to Murray and Wintle (2000) (Pg. 8, line 29) as essentially the preheat plateau is for the same purpose as in luminescence dating. I hope that this improves its clarity.*

*I have also added this sentence:*

[revised manuscript text omitted]